

# Weak integrability breaking and level spacing distribution

**D. Szász-Schagrin[1], B. Pozsgay [2] and G. Takács [1,3]**

**1** BME Momentum Statistical Field Theory Research Group, Department of Theoretical
Physics, Budapest University of Technology and Economics, Budapest, Hungary
**2** MTA-ELTE "Momentum" Integrable Quantum Dynamics Research Group,
Department of Theoretical Physics, Eötvös Loránd University, Budapest, Hungary
**3** MTA-BME Quantum Correlations Group (ELKH), Department of Theoretical Physics,
Budapest University of Technology and Economics, Budapest, Hungary

## Abstract

Recently it was suggested that certain perturbations of integrable spin chains lead to
a weak breaking of integrability in the sense that integrability is preserved at the first
order in the coupling. Here we examine this claim using level spacing distribution. We
find that the volume dependent crossover between integrable and chaotic level spacing
statistics which marks the onset of quantum chaotic behaviour, is markedly different for
weak vs. strong breaking of integrability. In particular, for the gapless case we find that
the crossover coupling as a function of the volume $L$ scales with a $1/L^2$ law for weak
breaking as opposed to the $1/L^3$ law previously found for the strong case.

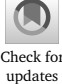

# 1 Introduction

Mean values of generalised currents in one-dimensional integrable models have attracted considerable interest lately, mainly due to the recent theory of Generalised Hydrodynamics which describes non-equilibrium dynamics at the Euler scale [1,2]. These currents express the continuity relation for conserved charges responsible for integrability, which can be exploited for a hydrodynamic description of the ballistic flow of the quasi-particles. This description is useful only as far as the exact mean values of the currents in local equilibrium conditions are known, and indeed in the thermodynamic limit a simple exact formula was postulated in [1,2], which was proven for relativistic quantum field theories [1,3], and for the spin current of the XXZ spin chain in [4], and the classical Toda chain in [5]. These exact results were extended to finite volume in [6] with a proof that applies to systems including Heisenberg spin chains, the Lieb-Liniger interacting Bose gas, and integrable quantum field theories with diagonal scattering. In [7] an algebraic construction was given for the current operators of the integrable spin chains, which led to an alternative rigorous proof of their mean values.

Recently [8] it was discovered that these exact results are connected to the so-called long range deformations of integrable spin chains which emerged in the context of the AdS/CFT correspondence [9–11]. These long range spin chains are obtained as a one-parameter deformation of ordinary short-range spin chains (such as the XXZ model) and preserve integrability to each finite order in the deformation parameter $g$, with the interaction range growing order-by-order. Strictly speaking, these deformations are only defined in infinite volume. In the work [12] it was demonstrated that these long range deformations have a deep connection to $T\bar{T}$-deformations of integrable quantum field theory [13,14] (see also [15,16]), by sharing the same algebraic origin which allowed the proof of factorisation for the expectation values of the operators which trigger the deformation of the spin chain.

It turns out that the existence of the exact formulae for the current expectation values is implied by the following observation made in [8]: for each generalised current there is a long range deformation such that the given current operator itself is the leading perturbing operator. This implies that the perturbation of the spin chain by the generalised current operator is integrable to the leading order in the deformation parameter $g$, but integrability is generally expected to be broken at higher orders since to maintain it necessitates the inclusion of progressively longer and longer range interaction terms at higher orders of $g$. We call this breaking of integrability 'weak' in distinction to 'strong' breaking of integrability which already happens at the first order of perturbation, and where integrability cannot be maintained by improving the perturbing operator order-by-order in $g$. At present it is believed that the only two possibilities for weak integrability breaking are given by the current- and $T\bar{T}$-like deformations, see [8–12].

The concept of weak integrability breaking is novel and it opens up new questions about the physical behaviour of integrable and nearly integrable models. Earlier works did not make a distinction between different forms of integrability breaking (for a recent review see [17])[1], and the physical consequences of the different types of perturbations started to emerge only recently. For example the work [18] studied integrability breaking within the framework of Generalised Hydrodynamics, where it was also found that the perturbations by the current operators do not break integrability at the leading order: while a generic perturbation with coupling $\lambda$ is expected to thermalise the system on a time scale of $T \sim \lambda^{-2}$, this effect is missing for the current operators on the same time scale. We also note the recent work [19] which argues that the perturbations with weak integrability breaking actually span the tangent

---

[1]We note that while Ref. [17] also uses the expression "weak integrability breaking", it is used in a very different way from the present work, since there it simply means perturbation with small coupling constant, and does not refer to a special class of perturbing operators.

of space of integrable models within the full space of local many-body models. This is an alternative explanation for the existence of weak integrability breaking: it arises as an effect of following the "tangent line" instead of remaining within the submanifold of integrable models, thus leading to much weaker effects than a perturbation which is "orthogonal" to the tangent space.

The onset of non-integrable a.k.a. chaotic behaviour in many-body systems can be investigated via the statistics of energy spectra [20–28]; in some cases even analytical results can be obtained [29]. For integrable systems the level spacing distribution is known to follow Poissonian statistics, while for non-integrable case random matrix theory predicts the so-called Wigner-Dyson distribution whose precise form depends on the relevant random matrix ensemble. In a finite volume, the transition between the two distributions is a smooth crossover, with the crossover coupling going to zero in the thermodynamic limit. For systems without a spectral gap, the crossover coupling scales as a (negative) power of the volume, with the exponent depending whether the local degrees of freedom are interacting or not [30]; for the case with interactions, the behaviour was determined to be $L^{-3}$, irrespective of the spatial dimensionality of the system.

In this short paper we set out to investigate the weak breaking of integrability by exploring the level spacing statistics for the case of the spin-1/2 XXZ spin chain. We compare a known strong integrability breaking term which introduces next-to-nearest-neighbour interactions to the weak integrability breaking perturbation provided by the lowest non-trivial generalised current. We start by presenting the Hamiltonian and the perturbations in Section 2, and then turn to the results for the level spacing statistics in Section 3, where we examine the crossover between the Poissonian distribution characteristic for integrable systems to the Wigner-Dyson distribution signalling the breaking of integrability, as a function of the integrability breaking coupling and the volume, and investigate the dependence of the crossover coupling on the volume. We present our conclusions in Section 4.

## 2 The Hamiltonian and its perturbations

### 2.1 The XXZ spin chain and the current operator

The spin-1/2 XXZ spin chain of length $L$ is defined by the Hamiltonian

$$H_{XXZ} = \sum_{i=1}^{L} \left[ s_i^x s_{i+1}^x + s_i^y s_{i+1}^y + \Delta s_i^z s_{i+1}^z \right], \tag{1}$$

where $s^j = \frac{1}{2}\sigma^j$ are the spin operators with $\sigma^j$ denoting the Pauli matrices

$$\sigma^x = \begin{bmatrix} 0 & 1 \\ 1 & 0 \end{bmatrix} \quad \sigma^y = \begin{bmatrix} 0 & -i \\ i & 0 \end{bmatrix} \quad \sigma^z = \begin{bmatrix} 1 & 0 \\ 0 & -1 \end{bmatrix} \tag{2}$$

and periodic boundary conditions $s_{L+1}^a \equiv s_1^a$. The model has three phases controlled by the anisotropy parameter $\Delta$: for $-1 < \Delta < 1$ the spectrum is gapless, while for $\Delta > 1$ and $\Delta < -1$ there is a non-zero gap. The two massive phases are physically different: for $\Delta < -1$ the system is in a ferromagnetic Ising phase, while $\Delta > 1$ corresponds to an antiferromagnetic Ising phase. The boundary points $\Delta = \pm 1$ are special as the $U(1)$ symmetry of the theory generated by the conserved charge

$$S^z = \sum_{i=1}^{L} s_i^z \tag{3}$$

is enhanced to $SU(2)$.

The XXZ chain is integrable: there exists a family of local conserved quantities

$$Q_n = \sum_{l=1}^{L} q_{n,l} \,, \tag{4}$$

where $n = 1, \ldots, L$ and the charge density is supported on exactly $n$ neighbouring sites. Their conservation can be expressed as a continuity equation

$$j_{n,l+1} - j_{n,l} = i \left[ q_{n,l}, H_{XXZ} \right] \,, \tag{5}$$

where $j_{n,l}$ is the conserved current corresponding to the charge $Q_n$.

The first two charges are $Q_1 = S^z$, and $Q_2$ which is nothing else but the Hamiltonian (1). These charges are related to the $U(1)$ symmetry and time translations, and are generally present in all the models considered in this work, including the non-integrable ones.

The first nontrivial charge corresponding to integrability of the XXZ Hamiltonian is

$$Q_3 = \sum_{l=1}^{L} q_{3,l} \tag{6}$$

$$q_{3,l} = s_{l-1}^x s_l^z s_{l+1}^y - s_{l-1}^y s_l^z s_{l+1}^x + \Delta \left( -s_{l-1}^z s_l^x s_{l+1}^y + s_{l-1}^z s_l^y s_{l+1}^x - s_{l-1}^x s_l^y s_{l+1}^z + s_{l-1}^y s_l^x s_{l+1}^z \right) \,,$$

with the corresponding current given by

$$
\begin{aligned}
j_{3,l} = -\frac{1}{2} \Big[ &2\Delta \big( s_{l-2}^x s_{l-1}^y s_l^x s_{l+1}^y + s_{l-2}^x s_{l-1}^z s_l^x s_{l+1}^z + s_{l-2}^y s_{l-1}^x s_l^y s_{l+1}^x + s_{l-2}^y s_{l-1}^z s_l^y s_{l+1}^z \\
&+ s_{l-2}^z s_{l-1}^x s_l^z s_{l+1}^x + s_{l-2}^z s_{l-1}^y s_l^z s_{l+1}^y - s_{l-2}^x s_{l-1}^y s_l^y s_{l+1}^x - s_{l-2}^y s_{l-1}^x s_l^x s_{l+1}^y \big) \\
&- 2 \big( s_{l-2}^x s_{l-1}^z s_l^z s_{l+1}^y + s_{l-2}^y s_{l-1}^z s_l^z s_{l+1}^y \big) - 2\Delta^2 \big( s_{l-2}^z s_{l-1}^x s_l^x s_{l+1}^z + s_{l-2}^z s_{l-1}^y s_l^y s_{l+1}^z \big) \\
&- \frac{1+\Delta^2}{4} \big( s_{l-1}^x s_l^x + s_{l-1}^y s_l^y \big) - \frac{\Delta}{2} s_{l-1}^z s_l^z \Big] \,.
\end{aligned}
$$

## 2.2 Perturbations, norms and effective coupling

We are interested in perturbing the XXZ chain by the current

$$H_J = H_{XXZ} + g_3 J \,, \tag{7}$$

where

$$J = \sum_{l=1}^{L} j_{3,l} \,, \tag{8}$$

which is supposed to break integrability only at higher order. As a benchmark to the strength of integrability breaking, we also consider a perturbation breaking integrability by the next-to-nearest-neighbor interaction (NNNI) term

$$\mathcal{O}_{NNNI} = \sum_{i=1}^{L} s_i^z s_{i+2}^z \,, \tag{9}$$

which leads to the Hamiltonian

$$H_{NNNI} = H_{XXZ} + g_N \sum_{i=1}^{L} s_i^z s_{i+2}^z = \sum_{i=1}^{L} \left[ s_i^x s_{i+1}^x + s_i^y s_{i+1}^y + \Delta s_i^z s_{i+1}^z + g_N s_i^z s_{i+2}^z \right] \,. \tag{10}$$

We remark that integrability breaking also affects transport properties, which was investigated in [31] exactly for the above NNNI perturbation of the spin-1/2 XXZ chain.

To compare the strength of integrability breaking, we introduce the effective coupling $g_{\text{eff}}$ of an operator $\mathcal{O}$ as

$$g_{\text{eff}} = g_{\mathcal{O}} n_{\mathcal{O}}, \tag{11}$$

where $g_{\mathcal{O}}$ is the coupling appearing in the Hamiltonian $H = H_0 + g_{\mathcal{O}} \mathcal{O}$, and $n$ is defined from the norm of the operator as the coefficient of the leading asymptotics

$$\|\mathcal{O}\|_2 = \sqrt{\text{Tr} \, \mathcal{O}^\dagger \mathcal{O}} \tag{12}$$

as the coefficient $n$ of the leading asymptotic term

$$\|\mathcal{O}\|_2 = n_{\mathcal{O}} L + \dots, \tag{13}$$

which is linear in the volume $L$ due to the expression of $\mathcal{O}$ as the translation invariant sum of localised terms (8,9). The advantage in parameterising the strength of the perturbation with $g_{\text{eff}}$ is that its value is invariant under a rescaling of the perturbing operator, and it also facilitates the comparison of the strengths of different perturbing operators.

The norm of $J$ depends on $\Delta$, while that of $\mathcal{O}_{NNNI}$ does not; some explicit values are shown in Table 1. Since the perturbing operators are given as translation invariant sums over localised interaction terms, their norm is expected to be extensive in the volume. Indeed, as illustrated in Fig. 1, their norms change linearly with the volume, apart from some fluctuations for $J$ which are due to the fact that $L$ is changed in steps of 2, while the one-site term $j_{3,l}$ is localised on 4 sites. The coefficient $n_{\mathcal{O}}$ can be computed by fitting a linear function to the norm values as a function of $L$, and is shown in the last column of Table 1.

Table 1: The norms of the perturbing operators $J$ and $\mathcal{O}_{NNNI}$.

| | $L$ | 10 | 12 | 14 | 16 | 18 | 20 | $n_{\mathcal{O}}$ |
|---|---|---|---|---|---|---|---|---|
| | $\Delta = 0.2$ | 0.742 | 1.306 | 1.232 | 1.841 | 1.772 | 2.346 | 0.143 |
| $\|J\|_2$ | $\Delta = -1.2$ | 2.376 | 3.115 | 3.357 | 4.614 | 4.726 | 6.023 | 0.348 |
| | $\Delta = 1.2$ | 2.274 | 3.115 | 3.455 | 4.614 | 4.412 | 6.023 | 0.340 |
| $\|\mathcal{O}_{NNNI}\|_2$ | | 1.500 | 2.000 | 2.500 | 3.000 | 3.500 | 4.000 | 0.250 |

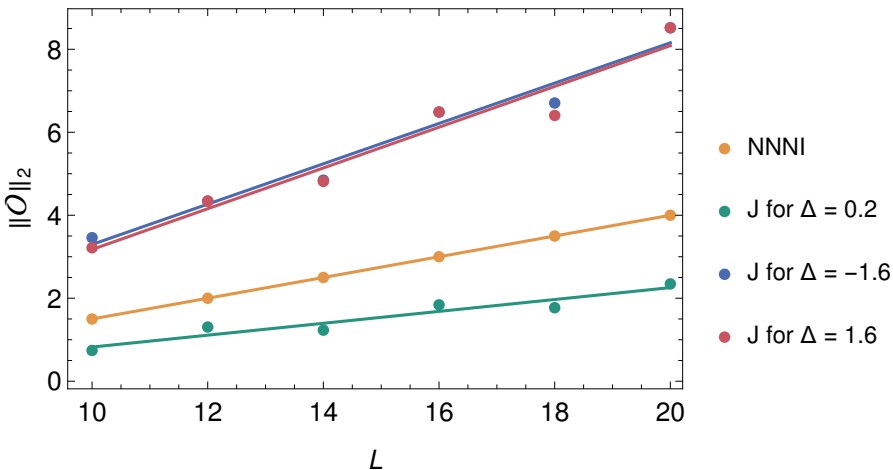

Figure 1: Norm of $J$ and $\mathcal{O}_{NNNI}$ as a function of the system size $L$ in the three phases.

# 3 Level spacing distribution in the XXZ chain and its perturbations

The level spacings of a given system described by a Hamiltonian $H$ with eigenvalues $\lambda_i$ are defined as the differences $S_i = \lambda_{i+1} - \lambda_i$ between eigenvalues ordered as a monotonically increasing sequence $\lambda_1 \leq \lambda_2 \leq \dots$. The distribution of normalised level spacings $s_i = S_i/\bar{S}$ (where $\bar{S}$ is the mean level spacing) is called the level spacing distribution $P(s)$. For integrable systems, the level spacing distribution is exponential:

$$P(s)_I = e^{-s},\qquad(14)$$

while for non-integrable systems it is given by the Wigner-Dyson distribution, which takes the following form for the orthogonal Gaussian ensemble[2]

$$P(s)_{NI} = \frac{\pi}{2}s e^{-\frac{\pi}{4}s^2}.\qquad(15)$$

Therefore, the level spacing distribution is an explicit indicator of integrability and its breaking. The above predictions follow from random matrix theory. For an integrable Hamiltonian, the different levels do not interact due to the presence of higher conserved charges, and so the distribution of eigenvalues is a Poissonian one, leading to exponential distribution of the normalised level spacings. Breaking integrability results in level repulsion, and so small level spacings are suppressed. When considering the spectrum as a function of a parameter such as volume, integrability implies that levels approaching each other as a function of volume simply cross, while in the non-integrable case they avoid each other due to level repulsion. In the limit of infinite matrix size, the level spacing distribution changes suddenly from exponential to Wigner-Dyson for any non-zero value of the integrability breaking coupling $g$.

For a spin chain of finite length, however, the Hilbert space is finite dimensional, and so the level spacing distribution is a continuous function of the coupling, with the transition becoming sharper for larger volumes [20,24]. In addition, when constructing the level spacing from the full spectrum it is found to deviate from the random matrix prediction due to the structure dictated by quasi-particle excitations, which is in turn due to the locality of the Hamiltonian. This problem can be solved by constructing the level spacing distribution from the middle part of the spectrum, for which we take the middle two-thirds of the computed levels. Furthermore, to get rid of degeneracies corresponding to trivial symmetries [32,33] the level spacing distribution is extracted from a sector with total momentum zero, even spatial parity and a fixed (non-zero) $S_z$ value[3]. In the examples below we present the results obtained for $S_z = 2$; similar results were obtained for $S_z = 1$ and 3.

## 3.1 The integrable XXZ chain

The level spacing statistics of the XXZ chain of length $L = 22$ in the different phases is shown in figure 2. It can be described very well by an exponential curve, with the overall normalisation of the distribution as the only fitting parameter.

---

[2]Due to the fact that the Hamiltonians considered here are real and symmetric, it is the orthogonal Gaussian ensemble which is relevant here.

[3]Note that the $S_z = 0$ sector has an additional spin-flip symmetry.

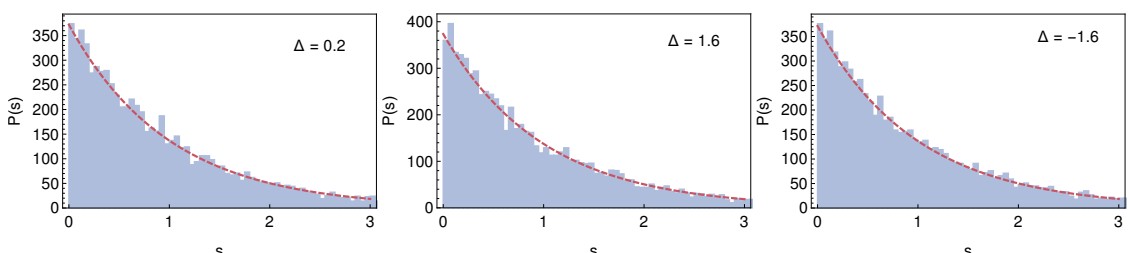

Figure 2: Level spacing statistics of the XXZ spin chain of length $L = 22$ for (left to right) $\Delta = 0.2, 1.6, -1.6$ with the exponential distribution fitted (dashed red line).

## 3.2 Perturbed system and crossover from Poisson to Wigner-Dyson statistics

Switching on a suitably large value for the integrability breaking coupling the level spacing distribution is changed into the Wigner-Dyson statistics characteristic of quantum chaos, as illustrated in Fig. 3. Again, the only fitting parameter is the overall normalisation of the distribution.

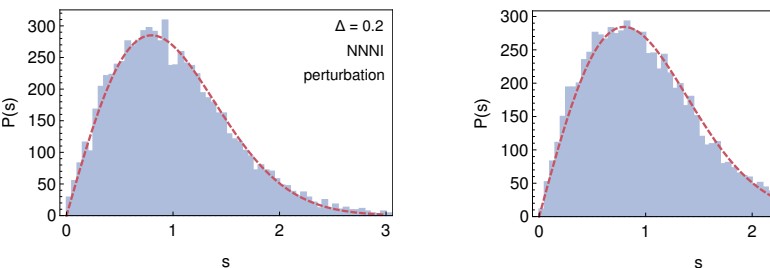

Figure 3: A typical level spacing distribution of a non-integrable system with the Wigner-Dyson distribution fitted (red dashed line). The left panel shows the case of $H_{NNNI}$ with $L = 22$ $\Delta = 0.2$ and $g_{\text{eff}} = 0.1$, while the right panel belongs to $H_J$ with $L = 22$, $\Delta = 0.2$ and $g_{\text{eff}} = 0.42$.

In a finite volume, varying the strength of the integrability breaking coupling leads to a crossover between the exponential and Wigner-Dyson distributions. The crossover can be quantified by determining the position of the maximum of the level-spacing distribution, which moves from the origin to the position $\sqrt{2/\pi}$ characteristic for the Wigner-Dyson distribution.

The normalised level spacings were sorted into bins of width 0.15 for $L = 16, 18, 20$, and of width 0.1 for $L = 22, 24$. The resulting distribution was then smoothed by applying a Gaussian filter of kernel radius $r$ to the raw histogram to suppress fluctuations due to finite bin size, with the choice $r = 6$ for lengths $L = 16$ and 18, and $r = 4$ for longer chains[4]. The determination of the position of the maximum of the level spacing distribution is illustrated in Fig. 4 for a chain of length $L = 22$.

---

[4]For chains with length smaller than 16 there are simply not enough level spacings to yield useful statistics.



(a) NNNI perturbation

(b) Current perturbation

Figure 4: Determining the dependence of the peak position on $g_{\text{eff}}$ for the NNNI (a) and current (b) perturbations for $L = 22$, $\Delta = 0.2$. The blue solid lines mark the result of the Gaussian filtering and the blue markers denote the extracted peak positions.

Note that the crossover happens at smaller couplings for the NNNI than for the current perturbation, indicating the difference in the 'strength' of integrability breaking; however, the decisive evidence eventually comes from the volume dependence considered in Subsection 3.4.

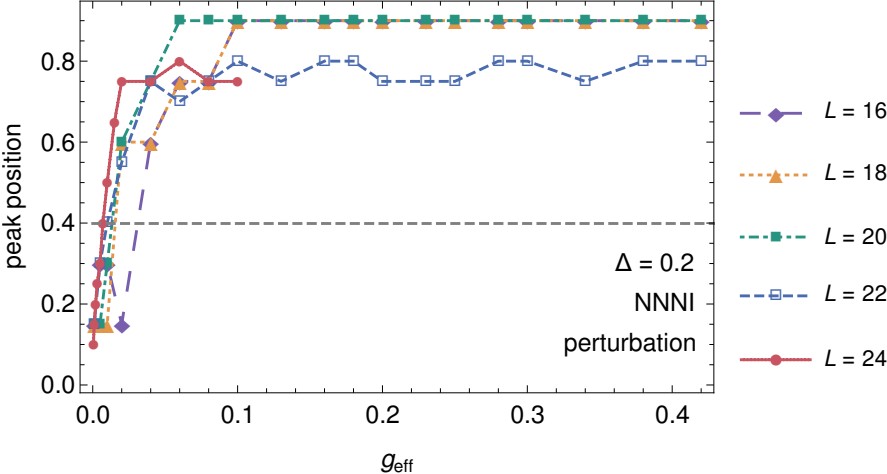

Figure 5: The position of the peak of the level spacing distribution corresponding to $H_{NNNI}$ for $\Delta = 0.2$ as a function of the effective coupling $g_{\text{eff}}$ for different chain lengths $L$. The dashed grey line marks $x_0$, where $2x_0 = \sqrt{2/\pi}$ is the position of the maximum of the exact Wigner-Dyson distribution. The transition from Poissonian to Wigner-Dyson statistics is faster for longer chains, as expected.

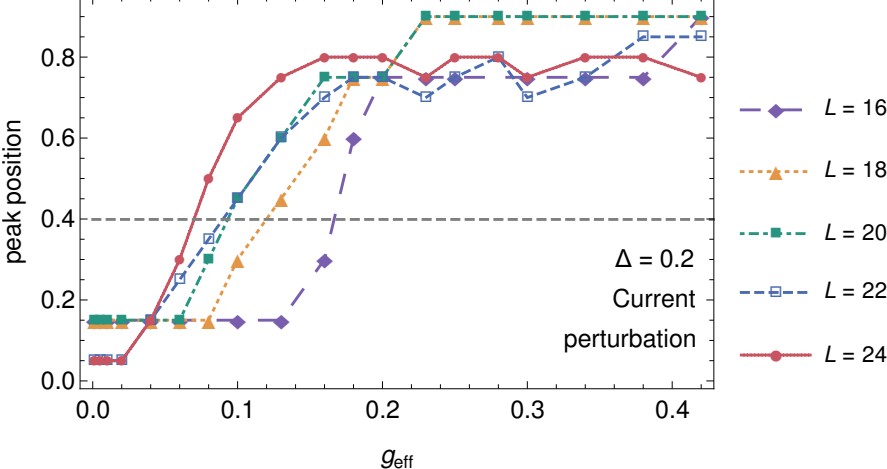

Figure 6: The position of the peak of the level spacing distribution corresponding to $H_J$ for $\Delta = 0.2$ as a function of the effective coupling $g_{\text{eff}}$ for different chain lengths $L$. The dashed grey line marks $x_0$, where $2x_0 = \sqrt{2/\pi}$ is the position of the maximum of the exact Wigner-Dyson distribution. The transition from Poissonian to Wigner-Dyson statistics is faster for longer chains, as expected.

Results for the NNNI perturbation in the gapless phase can be seen in Fig. 5, while for the current perturbation they are shown in Fig. 6. We remark that while the filtering facilitates the finding of the peak, the precision of its determined location is still limited by the bin size, leading to fluctuations in the determined peak positions which can be seen in the figures. As expected, the crossover occurs faster for longer chains, and the data also show that it is markedly slower for the current perturbation than for the NNNI case, supporting the idea that the current perturbation only breaks integrability at higher orders.

## 3.3 Integrability breaking and perturbation theory

To investigate the order of integrability breaking it is tempting to try and construct the level spacing distribution for the spectrum constructed from perturbation theory in the coupling $g$. However it turns out that this is not possible. Using simple first-order matrix perturbation theory to compute the spectrum of the perturbed Hamiltonian, and evaluating the resulting level spacing distribution demonstrates that it remains exponential even for the NNNI perturbation (9), as shown in Fig. 7.

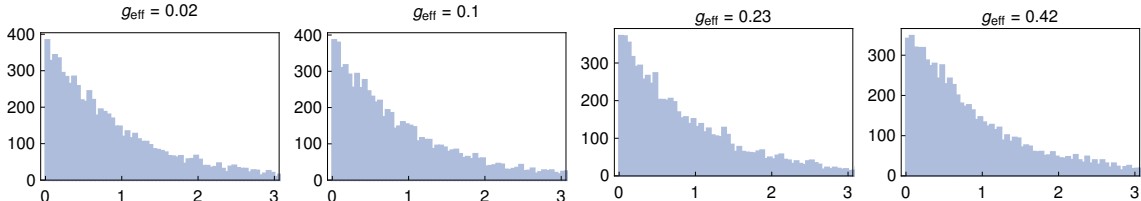

Figure 7: Level spacing statistics of $H_{NNNI}$ as computed from the spectrum obtained by first order perturbation theory for the case $L = 22$, $\Delta = 0.2$, for different values of the coupling.

To understand the reason, consider how a level crossing is lifted by a perturbation. In our model, the generic level crossing obtained as a function of the volume $L$ (continued to real values) happens between two levels. Let us consider an effective Hamiltonian description of a generic perturbation of the corresponding two level subsystem:

$$\begin{bmatrix} a_1(L) + g\Delta a_1(L) & g\epsilon(L) \\ g\epsilon(L) & a_2(L) - g\Delta a_2(L) \end{bmatrix}, \tag{16}$$

where the unperturbed energy levels are given by $a_{1,2}(L)$, and the functions $\Delta a_{1,2}(L)$ and $\epsilon(L)$ parameterise a general integrability breaking perturbation with strength $g$, in the two-level subspace. For the unperturbed Hamiltonian $g = 0$, the level crossing is located at the volume $L_0$ given by $a_1(L_0) = a_2(L_0)$; all that happens at first order is that the level crossing is shifted to a location given to the location $L_*$ which satisfies $a_1(L_*) + g\Delta a_1(L_*) = a_1(L_*) - g\Delta a_2(L_*)$, where $L_*$ can be computed as a series in $g$:

$$L_*(g) = L_0 - g\frac{\Delta a_1(L_0) - \Delta a_1(L_0)}{a_1'(L_0) - a_2'(L_0)} + O(g^2). \tag{17}$$

Therefore first order perturbation theory does not introduce a repulsion between levels, so the exponential distribution is unchanged.

When extending the perturbative calculation of the level spacing statistics to second order, the resulting energy levels turn out to be numerically unstable and no meaningful statistics can be constructed. For the two-level system above, an exact calculation of the energy levels shows that the off-diagonal term leads to level repulsion, which is responsible for changing the level spacing distribution. However, a perturbative evaluation of the two nearby energy levels results in

$$E_{1,2} = a_{1,2}(L) + g\Delta a_{1,2}(L) \pm g^2\frac{2\epsilon(L)^2}{a_1(L) - a_2(L)} + O(g^3), \tag{18}$$

which is unstable in the vicinity of a level crossing due to the presence of the energy difference denominator. As a result, the order of integrability breaking cannot be deduced by considering level spacing distribution of the spectrum obtained in perturbation theory in the coupling $g$.

## 3.4 Volume dependence of the crossover coupling

For the system with the integrability breaking turned on, let's define the crossover coupling $g_{cr}$ as the value of the effective coupling $g_{eff}$ for which the maximum of the intermediate distribution is at $x_0$, where $2x_0 = \sqrt{2/\pi}$ is the position of the maximum of the exact Wigner-Dyson distribution. The dependence of the crossover coupling on the volume $L$, $g_{cr}(L)$ is expected to be a monotonically decreasing function. More precisely, in the gapless phase it is expected to have a power-like behaviour $g_{cr} \propto L^{-\alpha}$ [30,34], while in the gapped phase finite volume corrections are expected to show exponential decay in the volume (cf. Subsection 3.4.2).

Note that since the transition is a smooth crossover, other definitions of $g_{cr}$ are also possible [30,34]; however, the finite size scaling is expected to be universal and therefore independent of these details. Indeed, in the following we recover the exponent $\alpha = 3$ obtained previously for the NNNI perturbation in the gapless phase [34].

### 3.4.1 Gapless phase

In the following we give results obtained by carrying out the above described method for the NNNI and current perturbations in the gapless phase, namely for $\Delta = 0.2$.

The obtained maximum positions can be seen in figure 8 along with the parabola $f(g)$ fitted to the resulting points around $x_0$. The crossover coupling is then obtained by solving $f(g_{cr}) = x_0$.

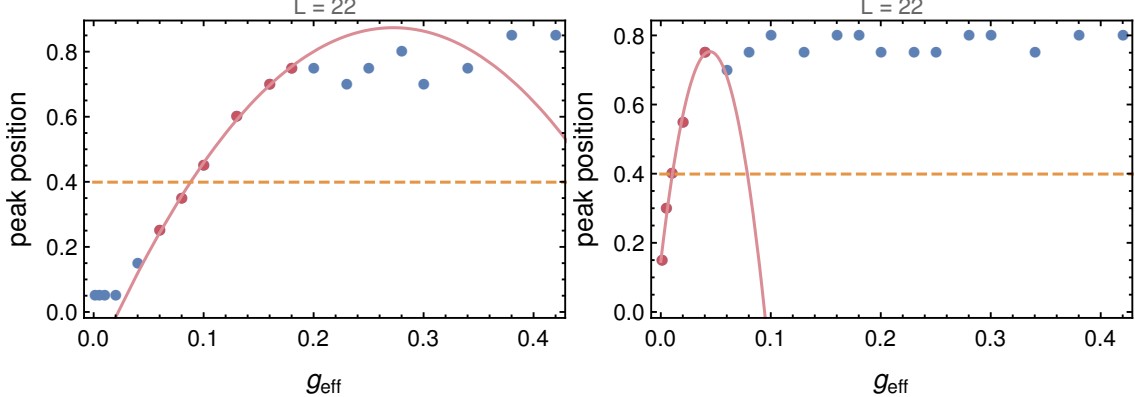

Figure 8: Obtaining $g_{cr}$ from the maximum positions for different effective couplings for $J$ (left) and NNNI perturbation (right) in the gapless phase ($\Delta = 0.2$). The red solid line is the parabola fitted to the red markers in the vicinity of $x_0$ (indicated by orange dashed line).

The $g_{cr}(L)$ functions obtained this way for both the NNNI and current perturbations can be seen in figure 9, and decay with a power of the volume $L$. Fitting a linear function $\log g_{cr}(L) = a + b \log L$ results in the exponents

$$b_J = -1.99 \pm 0.18 \,,$$
$$b_{NNNI} = -3.11 \pm 0.27 \,. \tag{19}$$

The value obtained for the NNNI case is consistent with the universal exponent $-3$ claimed for integrability breaking in a gapless chain [30,34], while the one obtained for the current perturbation is in agreement with the conjecture that the current perturbation breaks integrability at higher orders in perturbation theory.

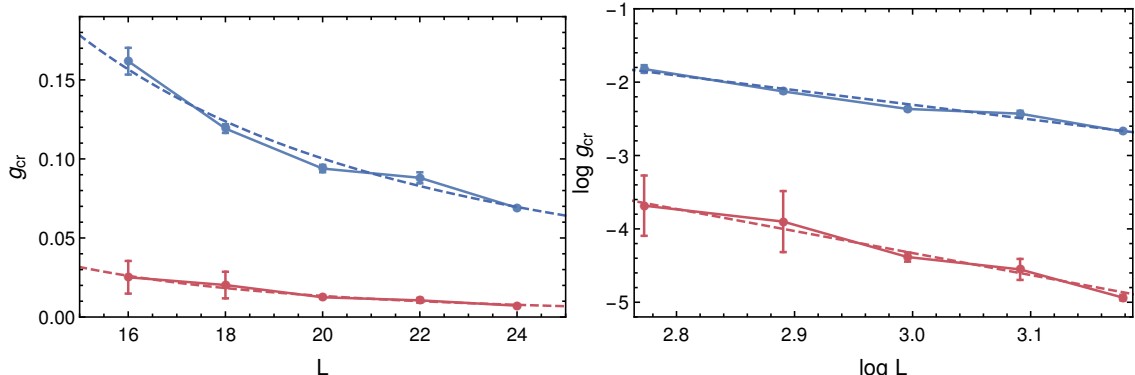

Figure 9: The crossover scale as a function of the volume for the current (blue markers) and NNNI perturbations (red markers) on normal (left) and log-log scale (right) as obtained in the gapless phase ($\Delta = 0.2$). Dashed lines correspond to the fitted $AL^{-3}$ (red) and $BL^{-2}$ (blue) respectively.

### 3.4.2 Gapped phase

In the gapped phase, finite size scaling is expected to be exponential in volume, since the presence of finite correlation length implies exponential decay of correlations. As a result, finite size effects on the spectrum (beyond its inevitable discretisation in finite volume) generally decay exponentially with the volume [35,36].

The crossover coupling as a function of the volume is shown for $\Delta = 1.6$ in Fig. 10, and for $\Delta = -1.6$ in Fig. 11. For the massive case, $\log g_{cr}(L)$ can be fitted with a linear function $c + dL$, with the following values for the coefficients $d$ for $\Delta = 1.6$:

$$
\begin{aligned}
d_J &= -0.056 \pm 0.014, \\
d_{NNNI} &= -0.157 \pm 0.020
\end{aligned}
\tag{20}
$$

and for $\Delta = -1.6$:

$$
\begin{aligned}
d_J &= -0.063 \pm 0.014, \\
d_{NNNI} &= -0.137 \pm 0.043.
\end{aligned}
\tag{21}
$$

Again we see a marked difference between the strength of integrability breaking for the two perturbations. The crossover coupling for the current perturbation in any given volume is an order of magnitude larger than for the NNNI perturbation, and the coefficient $d$ describing its decay with the volume is also significantly smaller, again in agreement with the conjecture that the current perturbation breaks integrability at higher orders in perturbation theory.

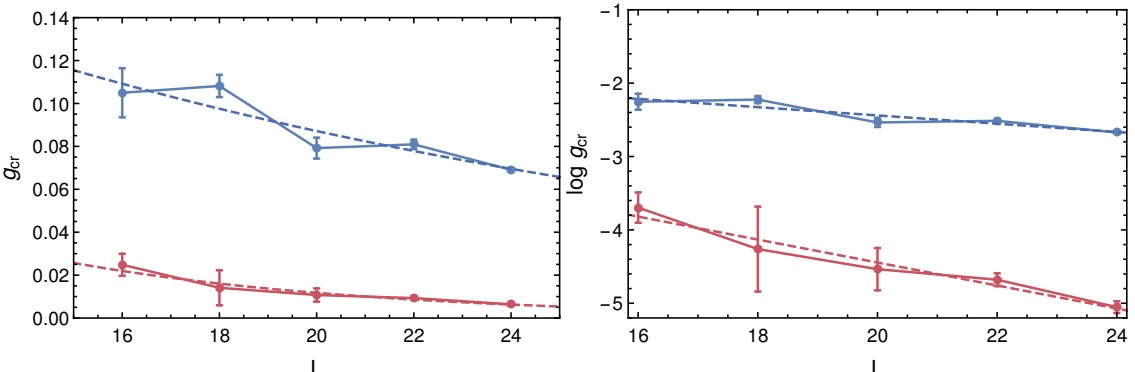

Figure 10: The crossover scale as a function of the volume for the current (blue markers) and NNNI perturbations (red markers) for $\Delta = 1.6$ (gapped phase, antiferromagnetic) on normal (left) and log scale (right). Dashed lines correspond to the fitted exponentials.

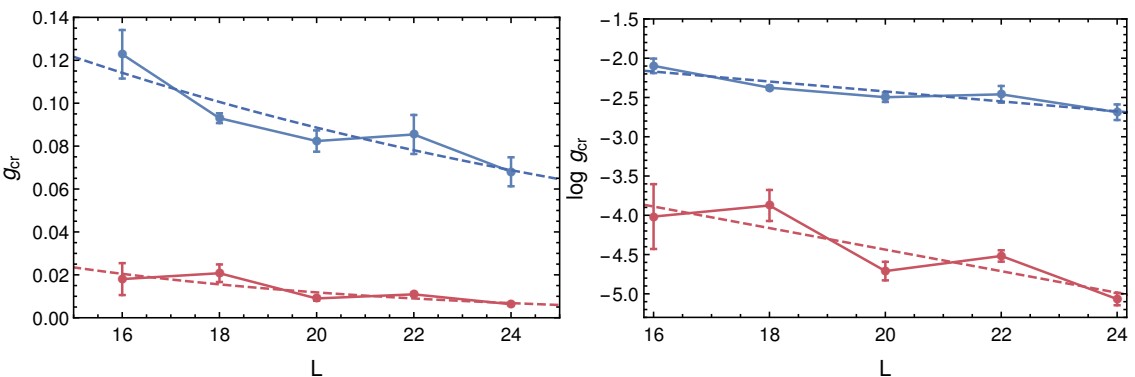

Figure 11: The crossover scale as a function of the volume for the current (blue markers) and NNNI perturbations (red markers) for $\Delta = -1.6$ (gapped phase, ferromagnetic) on normal (left) and log scale (right). Dashed lines correspond to the fitted exponentials.

## 4 Conclusions

In this work we examined the crossover from integrable to chaotic behaviour for weak breaking of integrability, defined as a perturbation of an integrable system which preserves integrability to first order in perturbation theory. As an example we took the XXZ spin chain perturbed by one of the currents that appear in the continuity equation for the higher conserved charges which imply integrability, and compared it to a usual ("strong") integrability breaking perturbation which was chosen to be a next-to-nearest-neighbour interaction term (NNNI).

The tool we used was the evaluation of level spacing distribution for a finite chain using exact diagonalisation. Since the extremal parts of the spectrum have special properties due to locality of the Hamiltonian, they were discarded with the middle two-thirds of levels kept. We then quantified the crossover between the integrable Poissonian and chaotic Wigner-Dyson statistics in the form of a crossover coupling. To facilitate the comparison between different operators, we rescaled their coupling constants by the operator norms (per unit volume), which also makes the values of the couplings independent of the choice of operator normalisation.

The behaviour of the crossover coupling as a function of the volume was found to be fully

consistent with the suggestion in [8] that the current perturbation only breaks integrability in higher order. For any fixed volume $L$ the crossover values of the rescaled couplings were significantly higher for the current perturbation compared to the NNNI case, and their decay with the volume was also significantly slower. In particular, for the gapless case while the crossover value of the NNNI perturbation follows the $1/L^3$ law found previously in [34], for the current perturbation we found a $1/L^2$ decay. In the massive regime the crossover coupling decreases exponentially with the volume, but again the decay for the current perturbation was significantly slower than for the NNNI. We note that this difference in the volume dependence is unaffected by the rescaling with the operator norms.

The slower finite-volume crossover between exponential and Dyson-Wigner distribution can be interpreted as a delayed onset of quantum chaos. The notion that there can be weaker and stronger versions of quantum chaotic behaviour has recently appeared in a different context related to out-of-time-ordered correlators [37]. However, the class of weak integrability breaking perturbations considered here define a concept of "weak quantum chaos" different from the one in [37], where instead it was related to the finite dimensionality of the local Hilbert spaces.

Since the $1/L^3$ law was claimed universal for strong integrability breaking in gapless models [30, 34] (albeit without analytic support), it is tempting to speculate that the exponent eventually depends on the order at which integrability is broken. However, as already known and also argued for in Subsection 3.3, perturbation theory cannot be applied to examine the crossover in the level spacing distribution if the level crossings resolved by the integrability breaking perturbation are generic i.e. involve only two levels. It is interesting to note that in quantum field theories obtained as perturbation of conformal field theories the crossover to the Wigner-Dyson behaviour takes place already in the perturbative regime [38]. The essential difference with the case considered here is that for the conformal spectrum the levels are generically multiply degenerate.

To sum up, our results strongly support the observations in [8,18] that perturbations of spin chains by generalised currents correspond to a weak integrability breaking which only happens at higher order in the perturbing coupling. An interesting question left open is to clarify the dependence of the finite volume crossover behaviour, especially the exponent appearing in the gapless case, on the order of integrability breaking.

### Acknowledgements

This work was partially supported by the National Research, Development and Innovation Office (NKFIH) under the research grant K-16 No. 119204, and also by the Fund TKP2020 IES (Grant No. BME-IE-NAT), under the auspices of the Ministry for Innovation and Technology. G. T. was also supported by the the National Research, Development and Innovation Office (NKFIH) via the Hungarian Quantum Technology National Excellence Program, project no. 2017-1.2.1-NKP- 2017-00001.

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
