# Peer review of "Weak integrability breaking and level spacing distribution"

_SciPost Physics, doi:SciPost Phys. 11, 037 (2021)_

## Round 2 · Referee Report · Anonymous (Referee 1) · 2021-4-7

Strengths
Weaknesses
Report
The authors investigated two types of perturbations in the XXZ model to make it non-integrable. The authors firstly numerically confirmed that the level spacing distribution in both the non-perturbing and strongly-perturbing regimes agree with the existed intuitions of integrable and non-integrable models. The authors then provided arguments to explain the intermediate regime where the exact form of distribution is unknown. Then the authors defined a critical coupling where the resulting peak of level spacing distribution matches with the non-integrable prediction. By assuming a power law scaling with the spin chain length, the authors obtained a linear fit for the "critical perturbation strength". The two types of perturbations behave differently, suggesting that one of them is special and "higher-order" in perturbation.
This work obtained interesting numerical results, and attempted to coherently summarize them. The difference between two perturbations is clear and therefore supporting existed results. The perturbation argument is original and seems working in understanding this problem.
However, the content of this manuscript does not meet the four expectations of SciPost Physics. The results are good and clear, but not groundbreaking to attract researchers to follow, nor creating new views to connect different areas.
Given this concern, I cannot recommend publication of this manuscript in its current form in SciPost Physics. However, I think if the authors can address the comments below, the manuscript should meet the criteria of SciPost Physics Core.
Major comments: - the current perturbation could be unique in terms of 1/L^2 scaling while others are 1/L^3. However, previous numerical evidences for the 1/L^3 are not about XXZ model. Could the authors provide more examples of perturbations to the XXZ models that exhibit similar scaling? - I think the terminology "critical strength" in a crossover is not appropriate. Maybe the authors can use "characteristic strength" instead. - why can one assume a power law for the scaling of "critical strength"?
Minor comments: - what's the behavior for lower L? I saw numerics mainly with L >= 16. Is there any significant finite-size effects with L < 16? I think with log(L) ranged from 2.75 to 3.2 it is not quite enough to perform a linear fit. - what's the purpose of showing both positive and negative Delta cases? Would one expect differences in the two? - earlier in the manuscript the authors mentioned cases of \Delta = +- 1.2 while later they investigated \Delta = +- 1.6. Is there any specific reasons for changing the values? (I think |\Delta|>1 is the gapped phase and therefore it makes no notable difference)

---

## Round 2 · Referee Report · Aaron Friedman (Referee 2) · 2021-4-23

Strengths
-
The authors clearly state the problem to be investigated in their work: namely they conjecture that perturbing an integrable spin chain by one of the model's conserved currents breaks the underlying integrability in a weaker sense than generic perturbations.
-
The authors' theoretical approach is sound: level statistics for integrable and chaotic systems have been studied at length and are a reliable diagnostic of integrability versus chaos.
-
The authors' numerical evidence is both thorough and convincing.
-
Integrability breaking is highly topical in the quantum dynamics and statistical mechanics communities.
-
The authors' abstract provides an exceptionally clear summary of the results.
-
The authors provide plenty of details to explain their methods and allow others to reproduce their results with ease. However, the numerical evidence is quite convincing and the analysis thereof thorough.
-
Despite the intimidating number of figures and their imposing size, they are quite easy to understand and support the authors' claims nicely. This paper can basically be understood from the abstract, figures, and a few equations, which I appreciate.
-
The paper's scientific merit is clear.
Weaknesses
-
The paper would benefit from revision throughout to make some of the language more clear. Many of the longer sentences are confusing, with ambiguous pronouns and dangling modifiers. Shortening sentences and clearly indicating "former" vs. "latter" or using punctuation to indicate clearly which statements go together (and general clarification) would be highly beneficial. This mostly applies to the introduction and conclusion.
-
The discussion of level statistics is quite pedagogical, e.g., while the introduction and background on integrability and integrability breaking assume a higher level of knowledge on the part of the reader than is reasonable. Even as someone with knowledge of integrability, I found that background on the models to which these results apply was lacking. The beginning of Sec. 3 is fine as written; my point here is that it would be nice if the introduction were written at a similar level.
-
The paper could more clearly contextualize the results in the context of the field at large. Who should care about this work and why?
As currently written, the paper provides a clear answer to a highly particular question; it is not clear that this will be of direct interest to more than a handful of people (namely, the authors Refs. 8-11). This can be remedied by providing more context for the results: Do the results have relevance to experiments? Do these results have implications for generalized hydrodynamics and integrability breaking? Is there any implication for chaos or other topics in quantum dynamics?
In the introduction, for example, the authors point out that using the GGE and GHD formalisms requires knowledge of the expectation values of infinitely many currents, but this is not possible in generic models. However, in the models with weakly broken integrability, this remains possible. Highlighting this claim and drawing a connection to ongoing work in closely related fields (e.g. quantum thermalization), could improve the paper's overall quality and appeal.
-
I would suggest the paper cite several recent works on integrability breaking, e.g. by R. Vasseur's group. However, as a coauthor of one of these works, I can't ethically insist on this point. However, the paper does benefit from the topical nature of integrability breaking and it would be reasonable to cite these works. Ideally, some discussion of how these results fit in with recent work on integrability breaking would improve the paper's quality and overall appeal.
-
I cannot say confidently that this paper meets one of the four expected Acceptance Criteria. However, with a bit of clarification and greater discussion of the results in the context of integrability more generally, I believe conditions 1 and 2 would be met. I think if the authors can make some of the proposed changes, this paper will be fine for SciPost.
-
It isn't clear if there is a "fit parameter" for the Poisson distribution. Naïvely, I would think e^{-s} has no fit parameter, but the use of the word "fitted" in the caption of one of Fig 2.
-
It might be helpful to state more clearly what is meant by weakly broken integrability. My takeaway was that weak breaking corresponds to a truncated version of the long range deformations that preserve integrability, where the truncation destroys the integrability, but only in a manner that preserves properties of the underlying integrable model.
-
Rather than saying that these perturbations "only break integrability at higher order", I would perhaps state that the "integrability is perturbatively stable to terms proportional to generalized currents". This is not a huge distinction, but seems like a safer claim since the paper only identifies the crossover strength, rather than a particular order (e.g. g^2) at which integrability breaks.
-
I found the discussion of the level crossing analytic confusing and had to read it several times. It would help to identify all the couplings (e.g. a1(L) ) that appear in the equations.
-
In the first paragraph of 3.4.1, a radius "r" is mentioned...what value is used?
Report
Overall, the research presented seems thorough; apart from some unclear sentences, all of the General Acceptance criteria are met.
What is slightly less clear to me is the extent to which the Expected Acceptance Criteria are met. I think by providing more context for their work as it relates to the field of integrability and quantum dynamics more broadly would remedy my concern.
If another referee does not share this concern, then I think it would be reasonable to accept the paper as is. Otherwise, I think with slight revision this paper ought to be accepted.
Requested changes
-
Please clarify the wording throughout. There are too many examples to list explicitly. In general, the longer sentences are difficult to understand or unclear.
-
Please clarify whether a fit parameter is used for the Poisson line in Fig 2, and the value of the radius r.
-
It would be helpful to comment on whether or not the same contrast between the J3 and NNNI perturbations is as stark if we use the naïve coupling, g, rather than g_{eff}.
-
Clarification of the variables in the equations in Sec. 3.3 would be useful. This discussion can be cleaned up somewhat as well.
-
Providing broader context for these results in the introduction and conclusion would greatly improve the paper in my opinion.

---

## Round 2 · Referee Report · Anonymous (Referee 3) · 2021-5-17

Strengths
1- Very timely topic 2-Theoretical predictions are put to numerical test 3-Convincing results
Weaknesses
2-Preparation of figures and captions is poor (see detailed comments)
Report
as they evolve from an integrable limit is a key goal in the field
of thermalization of closed quantum systems. The established knowledge on
the structure of local conserved quantities of integrable models combined
with the new developments from the theory of generalized hydrodynamics
provide important guidelines for this purpose. The present study
investigates the evolution of the level spacing distribution in the
vicinity of integrable points and puts theoretical predictions to a numerical
test. Central to this study is the recent observation of a connection between
so-called long-range deformations of integrable systems and generalized currents.
These generalized current give rise to only a weak breaking of integrability, with
consequences for thermalization times. A comparison of the effect of such a weak-integrability-breaking perturbation and a generic case is presented. The authors devise a way of extracting a critical coupling and find that the weak integrability-breaking perturbation requires much larger values than the generic perturbation to obtain Wigner-Dyson distributions, as reflected in a strikingly
different power-law dependence of the critical coupling strength.
The results and discussion are very interesting and the topic is timely. The manuscript should certainly be published in SciPost Physics, yet clarity of the presentation and in particular, quality of figures and captions need to be (substantially) improved.
Requested changes
1- There are older papers that pursued a similar spirit, see P. Jung, R. W. Helmes, and A. Rosch Phys. Rev. Lett. 96, 067202 (2006) and Antal et al. Phys. Rev. E 57, 5184 (1998), and Phys. Rev. E 59, 4912 (1999). The authors are encouraged to review this literature and add citations where appropriate.
2- One could point out that Q3 itself is a current for the XXZ chain, namely the energy current (see e.g., Zotos et al PRB 1997)
3- Figure 1: the y-axis label is unclear: A symbol lower case n was used earlier, what exactly is N?
4- Fig 1: the systematics from these data is unclear. The norms for the NNNI operator are neither systematically larger nor smaller than for the current. Please elaborate.
5- Figs 2 and 3: Please add axes labels. Legends in the individual panels would be very helpful (that is, stating what the value of Delta is for Fig. 2 and type of perturbation for Fig. 3).
6- What is the reason for selecting symmetry sectors with Sz > 0 (and not Sz=0) in the analysis of the level spacing distribution?
7- The qualitative discussion of Figs. 2--5 is very short and should be expanded.
8- Generally, I wonder why the authors do not extract the gap ratio and plot this as a function of L and geff. This could provide a much more compact and more accessible analysis of the data presented in Figs. 4 and 5.
9- Related to the previous point, the analysis of Sec. 3.4, which is the main part of the paper, concentrates on the evolution of the maximum of the distribution. What is the motivation to carry out this analysis and why is it justified to extract g_cr in this way? Does this lead to a unique value for g_cr? How does it compare to the approach of Ref. 30? Since this analysis is the core part of the present work, the authors should elaborate more on the reasoning. From the text, the definition sounds ad hoc.
10- Sec. 3.3: Typo: "remans"
11- Some more detail on how the results of Fig. 6 were obtained would be helpful. Is the observation that to first order, not much happens in the level spacing distribution, surprising? This seems generic as the example in Eq 16 shows.
12- Fig. 6, 7: What do the solid lines represent. This needs to be stated in the figure captions.
13- The notation "J"perturbation is potentially misleading as J (usually) is the coupling in the model. I suggest to just use "current perturbation".
14- Fig. 9 and 10: captions should state that these figures are for the gapless phase.
15- Type, page 11: "massivel"
16- Sec. 3.4.2: Why is the scaling expected to be exponential in the gapped phase? Please provide a reference at least.

---

## Round 3 · Referee Report · Aaron Friedman (Referee 2) · 2021-7-22

Strengths

The authors have satisfactorily addressed all of my concerns in my initial report. 1. The work is timely / topical 2. The presentation is clear and the numerical data are convincing 3. The abstract, introduction, and conclusion provide clear summaries of results 4. The authors explain how their results relate to other recent advances in integrability breaking and quantum chaos.

Weaknesses

The weaknesses I noted previously have been remedied.

Report

I recommend the current version of this paper for publication, and in my view it now meets/exceeds all SciPost acceptance/publication criteria .

The paper presents a convincing answer to an interesting and timely question, and relates these results to the field more broadly.

The paper is a very good fit for SciPost.

---

## Round 3 · Author Response

We sincerely thank the referees for their very helpful and constructive comments which were very useful in improving our presentation. We revised the paper accordingly, cf. the list of changes.

---

## Round 3 · List of Changes

Invited referee 1

Changes requested:

We included the paper by Jung et al. as we found it directly relevant to the issue of integrability breaking.

1. We have included the statement that Q3 is the energy current and inserted the appropriate citation.
2. We re-labelled the vertical axis in Figure 2.1 for clarification.
3. The systematics are explained in the text. Since the current explicitly depends on the anisotropy parameter Delta, its norm shows such a dependence as well, contrary to NNNI. In addition, we make it clear that these norms grow extensively with the volume, with some fluctuations due to the local extension of the individual terms. Further details are not so relevant since the norms themselves are dependent of the conventions in the definition of the operators. Indeed, this is one of the motivations of introducing the effective coupling, as stated now explicitly after (2.13).  
4. We added axes labels and the requested clarifications in Figs. 3.1 and 3.2.
5. Sector Sz=0 is invariant an additional symmetry under spin-flip (see footnote added on page 6), which would have been necessary to project out. It is possible to do that, but there is no added value in the data
6. We have expanded the discussion for two of the figures, now labelled 3.2 and 3.3. Old figures 4 and 5 have been replaced by 3.4 and 3.5, following suggestion 8.
7. We implemented the referee’s suggestion. To do that, we brought forward the description of our peak finding procedure from Subsection 3.4 to 3.2
8. We explain in Subsection 3.4 of the revised version that due to the smooth crossover there is no unique way to define where it happens. The position of the peak is very natural since it moves from the origin to the position of about 0.8. Previous works used other definitions. Note that our procedure is also justified by recovering the 1/L^3 scaling obtained in the previous works.
9. Typo corrected (together with some others we found in the text).
10. We added a more detailed description. Indeed, in the light of the detailed argument this is not surprising, so we reduced the figure (now 3.6) to just 4 subplots on one line. We still decided to keep it as it shows actual data and provides a visual illustration.
11. We removed the line from Fig 6 (now 3.6) as it has no role there. Figs 7 and 8 are now in Fig. 3.3 and include the description of the line and other clarifications.
12. We changed J perturbation to current perturbation everywhere.
13. We expanded the captions of the figures (now 3.7 and 3.8) as suggested.
14. Typo corrected.
15. We added a short explanation and references to the beginning of Subsection 3.4.2.

Invited referee 2

Changes requested:

1. We have extensively rewritten the text, paying attention to make the wording clear.
2. We made it clear that the only fit parameter both for exponential and Wigner-Dyson statistics is the overall normalization (which should match the total number of level spacings in the histograms).
3. The difference between the volume dependences is independent of the rescaling of the couplings. We added a sentence to the conclusions. The rescaling is primarily motivated by eliminating the dependence on multiplicative redefinition of the perturbing operators. It also makes their couplings more directly comparable, but not in a really strict sense. It is the volume dependence of the crossover coupling which really shows the difference in the strength of integrability breaking.   
4. We have added clarifications about the notations and hope it’s more readable now.
5. We have added significant new discussion to the introduction putting our results in context (the 4th paragraph). We also modified slightly the abstract and added material in the conclusions to relate our results to the topic of quantum chaos. As emphasized in the conclusions, what we find here is in a sense a “weak quantum chaos”. This phrase has already been used in the literature in connection with out-of-time-order correlators, but the sense we use it here is novel and refers to a distinct class of integrability breaking operators, as emphasized now in the introduction and the conclusions.

Weaknesses addressed:

1. We have gone through the paper to improve the text.
2. We included a new paragraph in the introduction.
3. Cf. point 5 above.
4. We agree with the referee that the work of Vassuer’s group is relevant, and included the appropriate reference ([17]).
5. –
6. Cf. point 2 above.
7. The new paragraph in the introduction includes a clarification of the concept of weak breaking of integrability.
8. Cf. point 7  above.
9. We included the description of the functions in the argument in Subsection 3.3.
10. The radius used for the Gaussian filter is now included in the text. Note that due to a rearrangement (referee 1, point 8) this text is now in Section 3.2.

Contributed review

Answers to major comments:

• This is indeed an interesting issue. We could in principle take the next current, but explicit construction of these currents is quite tedious and error prone. In fact, when we took the current J3 we had to correct typos in our sources which we did by explicitly checking its required properties. In GHD they are more of a theoretical concept than objects of explicit computations, although there has been great progress in giving an algebraic construction for them (https://arxiv.org/abs/2005.06242).

We have found that while in principle we know infinitely many such operators, explicit construction would require quite a large effort. However, we agree that this is an interesting problem for the future. Note that – at least to our knowledge – there is no theoretical explanation for the 1/L^3 law either. We think this is an interesting problem, as we state at the end of our conclusions. • We changed the terminology “critical strength” to “crossover coupling”. • Power law scaling in gapless systems was observed in previous literature. In fact, the 1/L^3 law was found quite universal. It is not a great surprise that finite size effects scale with a power when the correlation length is infinite. On the other hand, we found exponential decay in the gapped regime, which is not surprising given the finite correlation length (cf. the beginning of Subsection 3.4.2).

Answers to minor comments:

• There are simply not enough states for L<16 to construct a meaningful statistics (see the inserted footnote 4 on page 7). We agree that this is not optimal, but previous studies had this limitation as well and we found no way on improving this situation. 
• The Delta>1 and Delta<-1 phases are physically different (Ising antiferromagnet vs. Ising ferromagnet). We now clarify this after eqn. (2.2).
• We have a larger set of data (all showing the same effects), and we simply did not maintain consistency of our selections. Now we changed the coupling everywhere to the same values (-1.6, 0.2 and 1.6).

---

## Editorial Decision

published